# Geographical Expansion of Avian Metapneumovirus Subtype B: First Detection and Molecular Characterization of Avian Metapneumovirus Subtype B in US Poultry

**DOI:** 10.3390/v16040508

**Published:** 2024-03-26

**Authors:** Muhammad Luqman, Naveen Duhan, Gun Temeeyasen, Mohamed Selim, Sumit Jangra, Sunil Kumar Mor

**Affiliations:** Animal Disease Research and Diagnostic Laboratory, Department of Veterinary and Biomedical Sciences, College of Agriculture, Food & Environmental Sciences, South Dakota State University, Brookings, SD 57007, USA; muhammad.luqman@sdstate.edu (M.L.); naveen.duhan@outlook.com (N.D.); gun.temeeyasen@sdstate.edu (G.T.); mohamed.selim@sdstate.edu (M.S.); sumit.jangra@sdstate.edu (S.J.)

**Keywords:** Avian metapneumovirus, subtype-B, respiratory disease, poultry, NGS, phylogeny

## Abstract

Avian metapneumovirus (aMPV), classified within the *Pneumoviridae* family, wreaks havoc on poultry health. It typically causes upper respiratory tract and reproductive tract infections, mainly in turkeys, chickens, and ducks. Four subtypes of AMPV (A, B, C, D) and two unclassified subtypes have been identified, of which subtypes A and B are widely distributed across the world. In January 2024, an outbreak of severe respiratory disease occurred on turkey and chicken farms across different states in the US. Metagenomics sequencing of selected tissue and swab samples confirmed the presence of aMPV subtype B. Subsequently, all samples were screened using an aMPV subtype A and B multiplex real-time RT-PCR kit. Of the 221 farms, 124 (56%) were found to be positive for aMPV-B. All samples were negative for subtype A. Six whole genomes were assembled, five from turkeys and one from chickens; all six assembled genomes showed 99.29 to 99.98% nucleotide identity, indicating a clonal expansion event for aMPV-B within the country. In addition, all six sequences showed 97.74 to 98.58% nucleotide identity with previously reported subtype B sequences, e.g., VCO3/60616, Hungary/657/4, and BR/1890/E1/19. In comparison to these two reference strains, the study sequences showed unique 49–62 amino acid changes across the genome, with maximum changes in glycoprotein (G). One unique AA change from T (Threonine) to I (Isoleucine) at position 153 in G protein was reported only in the chicken aMPV sequence, which differentiated it from turkey sequences. The twelve unique AA changes along with change in polarity of the G protein may indicate that these unique changes played a role in the adaptation of this virus in the US poultry. This is the first documented report of aMPV subtype B in US poultry, highlighting the need for further investigations into its genotypic characterization, pathogenesis, and evolutionary dynamics.

## 1. Introduction

Avian metapneumovirus (aMPV) is a highly contagious air-borne pathogen which infects turkeys and chickens, causing turkey rhinotracheitis (TRT) and swollen head syndrome (SHS), respectively [1]. aMPV can infect Guinea fowl, pheasants, and ducks as well [2]. The virus is mainly associated with upper respiratory infections leading to clumping and loss of cilia, which predispose the birds to secondary bacterial pathogens resulting in severe respiratory signs, high morbidity, and mortality. aMPV can infect the reproductive system as well, resulting in a significant decrease in egg production [3].

The aMPV is a single-stranded, non-segmented, enveloped, and negative-sense RNA virus (~13.3–14 kb in size) in the family *Pneumoviridae*, genus *Metapneumovirus* [4]. The genome contains eight genes: 3′-Nucleoprotein (N), Phosphoprotein (P), Matrix (M), Fusion (F), Matrix 2 (M2), Small hydrophobic (SH), attachment (G), and large polymerase (L)-5′, in this order, with leader and tailer sequences at both ends: 3′-le–N–P–M–F–M2–SH–G–L–tr-5′ [5]. The L and P are non-structural proteins involved in genome replication, while the others code for nucleocapsid, matrix, and envelope structural proteins. The viral envelope is embedded with F (which promotes viral-to-cell membrane fusion), G (a major antigenic determinant involved in viral attachment), and SH (a viroporin involved in membrane permeability) proteins [6,7,8]. Proteins N, P, and L encapsulate the viral RNA to form the viral ribonucleoprotein complex (RNP), while the accessory proteins M2-2 and M2-1 are involved in viral genome replication and the 3′-leader/5′-trailer contains viral transcriptional promoters [7,8].

The attachment (G) and fusion (F) proteins are the major structural proteins, which exist on the envelope of the virus and are responsible for virus attachment and fusion to the host cells [9]. The attachment protein is a highly glycosylated type II membrane protein containing neutralizing epitopes, which are responsible for virus attachment to the cell membrane. Thus, the genotypic characterization and phylogenetic classification of the aMPV is mainly based on the nucleotide sequence of the G protein [10]. The F protein is a more conserved type I membrane protein, which is initially synthesized as a precursor (F0) protein and then cleaved into F1 and F2 subunits. The amino acid residues at the cleavage site can be used as a motif to distinguish among various aMPV subtypes. Changes in these amino acid residues result in failure of cleavage and subsequent decrease in virus infectivity [11]. The L gene is a polymerase enzyme responsible for virus virulence and viability, and is highly sensitive to mutation [12].

The aMPV was first reported in South Africa in 1970. Afterwards, it spread to several European countries, including the UK, France, Germany, Hungary, and Italy. In the meantime, another distinct aMPV, later identified as subtype B, was reported in several countries, causing serious economic losses in both turkeys and chicken [2]. In the U.S., the aMPV outbreak was not detected until 1996. The first case of aMPV was reported from commercial turkey farms in Colorado and Minnesota. Based on genetic diversity, this aMPV was named as subtype C [13]. Subsequently, it was detected in several turkey farms in different states as well as in wild birds [14]. Additionally, the C subtype was sporadically reported in pheasants and ducks in Korea, Italy, France, and China [15] and a one-time detection in Chinese commercial chickens [16]. Based on a retrospective study, a strain detected in 1985 from commercial turkeys in France was named as subtype D due to high genetic divergence from subtypes A and B [17]. Recently, two new subtypes have been discovered in North America, one in the black back gull [18] and the other in Monk parakeet chicks [19].

Subtypes A and B are considered a threat for the poultry industry because of their highly contagious nature and broad geographical distribution all over the world. The epidemiological studies performed in Europe have consistently reported the absence of subtype A, which was clearly surpassed by subtype B [2,20,21,22]. A recent phylodynamic study [2] reported that subtype B was able to spread rapidly in Western European countries, followed by Eastern countries. Whole genome sequences (WGS) of subtype B have been reported from different countries using advanced next generation sequencing (NGS) [8]. A recent study from South Korea reported six whole genomes of subtype B from chicken samples collected from live bird markets during 2019–2022. Kariithi et al. [8] reported whole genome sequence analysis of subtype B vaccine strains BR/1890/E1/19 (PL21, Nemovac; Boehringer Ingelheim Animal Health, Brazil) and BR/1891/E2/19 (1062; Hipraviar, France) along with that of the pathogenic field strain VCO3/60616. Goraichuk et al. [23] reported the whole genome sequence of the subtype B strain Hungary/657/4, which was isolated from a turkey in Hungary in 1989.

The present study reports further geographical expansion of subtype B with first-time detection in US poultry. Molecular detection and characterization based on whole genome sequencing is the scope of this study.

## 2. Materials and Methods

### 2.1. Clinical Cases

In January 2024, respiratory disease outbreaks were reported in chicken and turkey farms mainly in the eastern states, e.g., North Carolina and Virginia. To investigate the cause of this outbreak, necropsy of affected birds was performed in the field and swabs (nasal, tracheal and choanal, and cloacal swabs), trachea and nasal cleft/turbinate, and additional oviducts from breeders were submitted to the South Dakota State University Animal Disease Research and Diagnostic Laboratory (SDSU-ADRDL). Samples were received from 221 affected farms across 10 states; 197 were from turkey farms and 24 from chicken farms. Five to ten tissue samples or two pools of swab samples (11 swabs in each pool) in BHI were submitted from each flock. All samples were maintained at 4 °C during shipment to the laboratory and underwent initial processing steps, including pooling, aliquoting, and storage at −80 °C until further analysis.

### 2.2. Metagenomic Sequencing

The initial submissions (*n* = 39) from turkey and chicken farms were processed for next generation sequencing (NGS) to obtain a clear picture of the pathogens associated with the outbreak. The tissue samples were homogenized in a Stomacher by adding 1:10 phosphate buffer saline (PBS). The homogenized tissue samples and swab samples were centrifuged at 4 °C for 15 min at 3200× *g*. The supernatants were collected and processed for viral RNA extraction. The samples selected for NGS were further clarified by centrifugation at 6000× *g* for 5 min. The supernatants were incubated at 37 °C for 90 min with nuclease cocktail to degrade the unprotected DNA and RNA. The RNA was extracted using a QIAamp Viral RNA Mini Kit (Qiagen, Hilden, Germany) without Carrier RNA in the lysis buffer AVL. The extracted RNA samples were processed for avian-specific host ribosomal RNA depletion [24]. The cDNA synthesis was performed using SuperScript III First Strand Synthesis System (Invitrogen, Carlsbad, CA, USA) with random hexamers FR26RV-N followed by double-stranded DNA amplification with Sequenase Version 2.0 DNA Polymerase (Applied Biosystems, Vilnius, Lithuania). PCR amplification of dsDNA was performed using TaKaRa rTaq with primers FR20RV [25]. The amplified dsDNA was purified, quantified, and used for DNA library preparation using Illumina DNA Prep kit (Illumina, San Diego, CA, USA). The libraries were loaded on Illumina MiSeq for 300 cycles sequencing.

### 2.3. Realtime RT-PCR

The extracted RNA from all samples was screened using a RealPCR AMPV A/B Multiplex RNA Mix kit (IDEXX, Montpellier, France) for detection of the RNA of AMPV/A and B (IDEXX-99-56487). The primers and probes in this RNA mix are designed for the identification of and differentiation between AMPV/A and B. TaqMan™ Fast Virus 1-Step Master Mix (Applied Biosystems-4444434, Vilnius, Lithuania), standardized at a total reaction volume of 20 µL, was used to optimize real-time RT-PCR. Each reaction contained 5 µL extracted RNA, 5 µL master mix, 5 µL primer and probe kit, and 5 µL RNase-free water. A total of 40 cycles were conducted in a 7500 fast Thermocycling (Applied Biosystems) under the following conditions: 5 min reverse transcription at 50 °C followed by an initial denaturation step at 95 °C for 20 s and 40 cycles of final denaturation at 95 °C for 3 s followed by an annealing step at 60 °C for 30 s. Selected chicken and turkey samples of different Ct values and from different states were submitted to the National Veterinary Services Laboratories (NVSL), Ames, IA for further confirmation.

### 2.4. Sequence Assembly and Annotations

The raw reads were processed with Kraken2 v2.0.8 [26] for k-mer based taxonomic classification of raw reads. To ensure data quality, reads were subsequently analyzed with FASTQC. Subsequently, low-complexity sequences, short reads (<50 bp, specify length), and adapter contamination were removed using Trimmomatic [27]. The final cleaned reads were aligned to the reference genomes of both turkey and chicken downloaded from (https://ensembl.org accessed on 10 January 2024) using STAR aligner [28]. All these steps were performed simultaneously with pySeqRNA [29]. The unmapped reads, potentially representing non-avian or viral sequences, were extracted from BAM files using SAMtools [30]. *De novo* assembly of these unmapped reads was performed using MegaHIT [31] with the parameters “-no-mercy” and k-mer sizes ranging from 21 to 141, with the aim of maximizing the assembly of diverse viral sequences. The resulting contigs were then subjected to two rounds of annotation for functional and taxonomic insights. Kraken2 was utilized again for rapid k-mer based taxonomic classification, providing preliminary identification of potential viral sequences within the assembled contigs. Subsequently, National Center for Biotechnology Information (NCBI) BLAST [32] was employed for more in-depth annotation, allowing for specific gene identification and comparison with known viral references. Further, the unmapped reads were mapped on the identified reference genome using BWA [33]. The resulting alignments, stored in BAM files, were subsequently analyzed with Reditools2 [34] to identify single nucleotide variations (SNVs) within the mapped reads, including both substitutions and indels.

### 2.5. Phylogenetic Analysis

Phylogenetic analysis of assembled aMPV genomes was performed with aMPV sequences available in GenBank. Multiple sequence alignment was performed using Muscle [35] and to quantify the degree of sequence similarity between the aMPV strain and other aMPV lineages, a percent identity matrix was generated using a custom Python script. The maximum-likelihood tree was generated using PhyML with the JC69 substitution model [36] and 100 bootstraps. Finally, the tree was plotted using a custom Python script.

## 3. Results

### 3.1. Clinical Cases

Affected turkeys showed nasal discharge, frothy eyes, and conjunctivitis followed in later stages by mucopurulent turbid nasal discharge, plugged nostrils, swollen infraorbital sinuses, rhinitis with a snick, sinusitis (white caseous mucous in the sinuses), large amounts of mucus in the tracheas, pericarditis, and air sacculitis (Figure 1A). Typical signs in chickens were decreased feed intake and huddling at about 5 weeks of age followed by the appearance of upper respiratory noise (snick, cough). The eyes of affected chickens became squinty and reddened, and showed noticeable swelling around the eyes (Figure 1B). Mortality gradually increased mainly due to secondary bacterial infection with *Escherichia coli* and *Ornithobacterium rhinotracheale* (ORT). Necropsy showed extensive thoracic and abdominal air sacculitis, pericarditis, perihepatitis, and generalized polyserositis (personal communication with field veterinarians). Affected turkeys, mainly 6–10 weeks of age, experienced 30% to 50% mortality. One of the first cases from North Carolina lost 80% of the birds in a barn. Chickens were mainly affected at about five weeks of age (personal communication with field veterinarians). Breeder flocks experienced a significant drop in egg production with and without respiratory signs.

### 3.2. Metagenomic Sequencing

The metagenomic sequence data analysis confirmed the presence of aMPV. The aMPV subtype B whole genome was assembled in six samples with and without other viruses (such as paramyxovirus and adenovirus) and bacteria (such as *E. coli* and ORT). This information was helpful in implementing commercially available real-time RT-PCR for screening of outbreak samples.

### 3.3. De Novo Genome Assembly and Annotation

High-throughput paired-end sequencing was conducted on the isolated RNA using the Illumina MiSeq platform, generating approximately 174 million reads (detailed read distribution available in Appendix A). This sequencing effort resulted in the assembly of six complete aMPV genomes, with five originating from turkey samples (ADRDL-1-5) and one from a chicken (ADRDL-6) sample. Genome sequences [13,508 nucleotides (NT) long] were assembled with coverage ranging from 384.25X to 3552.28X, indicating high-quality sequencing depth confidence in the assembled sequences. The full-length genome sequences were deposited in GenBank (accession numbers PP273456-PP273461).

### 3.4. Realtime RT-PCR

Samples from 221 farms in 10 states were tested by aMPV RT-PCR. A total of 197 turkey farms were tested from the following states: Virginia (74), North Carolina (97), Illinois (1), Iowa (2), Minnesota (7), Missouri (3), Pennsylvania (10), Indiana (1), and Wisconsin (2). Additionally, 24 chicken farms were tested, originating from North Carolina (23) and South Carolina (1). A total of 100 (50.76%) turkey farms were positive for subtype B, with Ct values ranging from 14.78 to 35. All 24 (100%) chicken farms tested positive for aMPV subtype B, with Ct values ranging from 17.61 to 34.76. Samples tested positive from three states (NC, PA, and VA). The geographical distribution of positive and negative turkey samples across farms is visually represented in Figure 2. All samples were negative for subtype A by our multiplex PCR.

Most turkey samples tested positive at 6 to 10 weeks, while chicken samples were positive at about 5 weeks of age. Positive samples were found in breeder farms as well (both turkey and chicken). Choanal cleft/choanal swab was the most common sample type collected from infected farms, representing about 88% of the total number of positive aMPV subtype B samples. The samples submitted to NVSL were confirmed positive for subtype B by aMPV subtype specific real-time RT-PCR and sequencing.

### 3.5. Phylogenetic Analysis

Phylogenetic analysis of the assembled genomes was performed using previously published sequences of all aMPV subtypes from GenBank (Table 1). The phylogenetic analysis based on WGS, G, and F sequences clustered all six sequences with subtype B sequences of field strains and vaccine strains reported from Europe and South America followed by subtype B sequences from Korea and China (Figure 3, Figure 4 and Figure 5). All six sequences clustered with 99.29 to 99.98% nucleotide identity, indicating that a single strain is circulating in both chicken and turkey populations in US. All six sequences showed 97.74 to 98.58% nucleotide identity with previously reported subtype B sequences, mainly VCO3/60616, Hungary/657/4, and BR/1890/E1/19. All six sequences showed significantly lower identities with aMPV/C (54%) and human metapneumovirus (hMPV, 54%). aMPV/A and D showed moderate identities (around 67%).

The highly variable attachment (G) protein showed maximum divergence (93–97% NT and 93–95% AA identities) with previously reported subtype B sequences. The G protein was highly divergent from aMPV/A (52% NT and 32% AA identities), subtype D (39% NT and 25% AA identities), and subtype C (25% NT and 15% AA identities). Interestingly, the fusion protein analysis revealed an even closer relationship with aMPV/Bs, showcasing a remarkable 99% identity with subtype B as compared to only 67% with aMPV/Cs (Table 2).

### 3.6. Single Nucleotide Variation (SNV) in Newly Detected aMPV/Bs and Other Subtype B Strains

Comparative analysis of the assembled aMPV subtype B genomes revealed intriguing differences compared to the reference strains VCO3/60616 and BR/1890/E1/19 (PL21). Mapping the raw reads identified 189–207 SNVs with VCO3/60616 and 206–220 SNVs with BR/1890/E1/19, with over 20% exceeding the 25% read frequency threshold. Notably, 23–30% of these SNVs were missense variants, potentially altering amino acid sequences. Interestingly, most missense mutations (21–24) clustered within the attachment G protein gene across both reference strains (Figure 6). This translates to a potential change in 3–4% of G protein amino acid residues, raising questions about potential functional implications. One unique AA change from T (Threonine) to I (Isoleucine) at 153 was reported only in chicken ADRDL-6 sequence. Hence, there was a change in polarity from neutral to hydrophobic aliphatic AA in the ADRDL-6 sequence. At position 322, there was a change from I to T in chicken the ADRDL-6 sequence, which was different from all five aMPV subtype B sequences from turkey samples (Figure 6). While this T AA is present in all reference sequences, I was detected in ADRDL-1–5 sequences from turkey samples; this AA change at the 322 position confirms the change in polarity in the ADRDL-1–5 sequences from turkey samples (Figure 6). There were twelve unique AA changes observed in G protein of ADRDL-1–6 subtype B sequences, with change in polarity at some positions, such as a change from acidic glutamic acid (E) to basic lysine (K) at position 172, strongly basic arginine (R) to neutral serine (S) at position 256, and neutral asparagine (N) to acidic aspartic acid (D) at position 346 (Figure 6). The missense mutations were detected in different proteins; details of these mutations are presented in Appendix A.

## 4. Discussion

This study is the first documented report of aMPV subtype B infection in US poultry. aMPV infection primarily manifests as a respiratory syndrome localized in the upper respiratory tract of birds, mainly turkeys and chicken. However, co-infection with opportunistic pathogens can exacerbate disease severity by exploiting immune suppression induced by aMPV. While aMPV infection may cause 100% morbidity, mortality varies significantly, ranging from 0 to 30%. In breeders and laying birds, aMPV infection can extend to the reproductive organs, resulting in significant economic losses due to decreased egg production and quality [1].

Six aMPV subtype B whole genomes were assembled from turkey (*n* = 5) and chicken (*n* = 1) samples from Virginia, and North Carolina. This finding was initially not anticipated, considering that only aMPV subtype C has previously been reported in the US turkey population [14]. Detection and characterization of emerging pathogens such as aMPVs can be performed rapidly using advanced NGS technologies, which can identify divergent strains that are not detected by conventional PCR and enable immediate outbreak response [8,18,19,37]. As a result of the early detection and confirmation of aMPV subtype B in this study, molecular and serological testing was implemented very quickly for further testing of new samples.

Of the 221 farms, 124 were positive for aMPV-B, with Ct values ranging from 17 to 35. This indicates rapid spread of this virus in a relatively short amount of time. Most of the infected birds from commercial turkey farms were less than 10 weeks of age, with a high percentage of positive results in the 6–10 weeks age group. This finding is in agreement with previous studies which have reported that while aMPV subtype B infects both turkey and chicken with up to 100% morbidity in all ages, birds in the age group from 4 to 9 weeks are the most susceptible to infection [18,19]. Moreover, although some positive aMPV/B PCR results were observed in tracheas, tracheal swabs, cloacal and tracheal swabs, oropharyngeal swabs, and oviducts, most of the positive results were from choanal cleft/choanal swabs. It has been reported that the upper respiratory tracts of both turkey and chickens are the predilection sites for aMPV replication. Therefore, the virus can be detected in the nasal discharge, choanal swabs, and scraping or swabbing of the turbinate and choanal clefts, but within the first 5 to 6 days from the onset of infection [38].

Previously, aMPV subtype C was the only documented subtype circulating in the US, primarily confined to sporadic outbreaks on specific commercial turkey farms within a limited geographical area of Upper Midwest, including Minnesota, Wisconsin, Iowa, and Colorado. Additionally, the reported spread and severity of infection were low, particularly in breeder flocks [39]. Consequently, the economic impact of aMPV/C in the US was considered minimal, and control measures primarily focused on implementing robust biosecurity practices, with limited vaccine use in breeder flocks. In contrast to aMPV/C, which historically exhibited limited geographic distribution, aMPV/B has demonstrated a significant increase in prevalence and broader geographic spread across Europe, Asia, the Middle East, and South America. This geographical expansion has resulted in substantial economic losses to both turkey and chicken farms. However, the factors driving this rapid and widespread emergence of aMPV/B remain unclear. While epidemiological data are currently insufficient to fully explain this phenomenon, ongoing research to elucidate the potential mechanisms and contributing factors is important [10].

aMPV subtype B was detected in both chicken and turkeys in the eastern United States, mainly in NC, PA, and VA, which is a different geographical area from the Upper Midwest where aMPV subtype C has been reported. These outbreaks were reported mainly from densely populated areas with both chicken and turkey farms. These states are on the Atlantic flyway, which may indicate the involvement of migratory birds in the introduction of subtype B into US poultry. However, there is no documented record of subtype B being detected in wild birds. A molecular epidemiology study from Italy reported that all wild bird samples were negative for subtypes A and B [40].

Remarkably, a high proportion of positive samples with low Ct values were observed in the eastern states, where both turkey and chicken farms are in the same proximity. On the other hand, most of negative samples or very high Ct suspected samples originated from farms in upper Midwestern states such as WI, MO, MN, IA, and IL where there is no close coexistence of turkey and chicken farms. This finding is in agreement with Tucciarone et al. [20], who stated that chickens have a significant role in the maintenance of aMPV/B infection by increasing virus circulation and diffusion without any evidence of host specificity between turkey and chicken. However, further challenge studies should be conducted to confirm the role of chickens in maintaining and spreading aMPV/B.

The 99.29 to 99.98% nucleotide identity in the study sequences indicates clonal expansion of a single strain. However, one unique AA change from T (Threonine) to I (Isoleucine) at 153 was reported only in the chicken aMPV sequence, which differentiated it from the aMPV subtype B sequences from turkey samples. The significance of this AA change is not known currently. Based on WGS analysis, the ADRDL 1–6 sequences reported in this study were 97.74 to 98.58% identical with previously reported subtype B sequences. This is interesting for the first introduction of this subtype B in US poultry with ~2% difference in the genome from previously reported sequences. However, based on G protein, which is the major antigenic protein, study sequences showed only 95% identity with previously reported sequences. The twelve unique AA changes with change in polarity in G protein of the study sequences may highlight that these unique changes played a role in the adaptation of this virus in US poultry. These data will be helpful in the development of new vaccines against these strains, as they are genetically different from currently available commercial vaccines.

There are contradictory reports on the evolution of aMPV, with some studies reporting that aMPV is a relatively slow-evolving virus when compared to other avian RNA viruses and others estimating that its rate of viral evolution is within the normal range [41,42,43]. However, viral evolution is based on both the pressure exerted by vaccine programs and on the type of host and environment; therefore multiple strains of the same subtype can phenotypically circulate in different portions of the world [5]. The genetic diversity in the study sequences highlights the urgent need to carry out more whole genome sequencing of this virus in order to better understand the variants circulating in the field as well as to understand the evolution of this virus over time.

The observed high degree of sequence identity (>99%) among the six newly detected aMPV strains, including one from a chicken farm, aligns with reports of aMPV/B infections in Europe, where the same strain has been isolated from both turkey and chicken farms. This suggests a lack of strict host specificity for the identified aMPV/B strain [10,20]. However, it is important to acknowledge that the current analysis is based on a limited number of sequences (*n* = 6). Further investigation with a broader set of samples from diverse geographical and host origins is necessary to definitively assess the host range and potential host-specific adaptations of this aMPV/B strain.

Franzo et al. [2] carried out phylodynamic analysis of subtype B around Europe and reported a high evolutionary rate of this virus, leading to the establishment of genetically and phenotypically different clusters among eastern and western countries. Hence, these clusters could affect the efficacy of natural or vaccine-induced immunity, and should be accounted for when planning control measure implementation. In addition, it is important to continue whole genome sequencing of positive samples from chicken and turkey at regular intervals to keep track of virus evolution in US poultry. Franzo et al. [2] reported significant strain exchange among turkey, guinea fowl, and chicken without any evidence of differential selective pressure or specific amino-acid mutations, suggesting that no host adaptation is occurring. This finding correlates with our results, where the WGS of subtype B from chicken and turkey samples were >99% identical.

Cecchinato et al. [10] studied fourteen sequences of aMPV subtype B isolates collected in Europe from the Veneto region of Italy between 1986 (eight strains from 1986–1994) and 2007 (six strains from 2001–2007), mainly to determine the effect of the commercial subtype B vaccine widely used in Italy. A single spray vaccination of one-day-old turkey poults with live attenuated vaccine was standard for commercial turkeys. This study reported eighteen and three AA substitutions in the 2004 and 1987 strains, respectively, in comparison with the vaccine. The emergence of new variants was attributed to selection pressure from vaccine-induced immunity. The same study reported changes in the charged amino acids K, R, D, E, and H and the glycosylation AAs S and T. This correlates with findings from the current study, where the maximum AA changes were detected in G protein, with most of the AA changes in charged AAs. Catelli et al. [44] performed a challenge experiment and concluded that vaccination was only effective in protecting against challenge by the 1987 strain; poor protection was observed in birds challenged with the 2004 strain. This indicates that these AAs in G proteins play a role in pathogenicity and antibody response.

Further investigation into these specific changes and their impact on G protein function and vaccine efficacy is crucial. This initial analysis highlights intriguing genomic variations in our aMPV-B isolates, particularly within the attachment G protein. Delving deeper into these findings through functional studies is essential to understanding their potential biological significance and impact on vaccine effectiveness.

## 5. Conclusions

This study has highlighted the first detection and molecular characterization of aMPV subtype B infection in US poultry, affecting both commercial turkeys and chickens across various age groups. The use of NGS was useful for early detection and whole genome sequencing of emerging subtype B strains in US poultry. Although the study sequences are closely related to previously reported sequences from Europe, unique AA changes were detected in the study sequences. Real-time RT-PCR is helpful in screening samples from the affected areas. However, further investigations remain important in order to better understand the virus’s host range, transmission dynamics, and potential adaptations in US poultry.

## Figures and Tables

**Figure 1 viruses-16-00508-f001:**
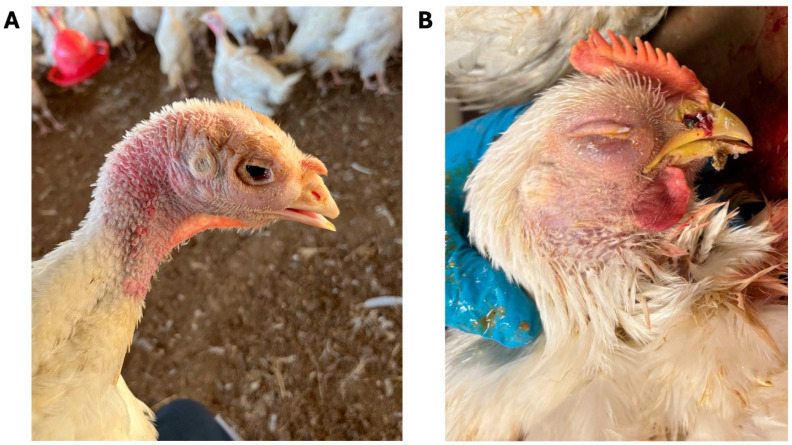
(**A**) Turkey showing swollen infraorbital sinuses, ocular discharge, and conjunctivitis (courtesy of Dr. Ashley Mason); (**B**) five-week-old chicken showing swollen head, swollen infraorbital sinuses, plugged nostrils, and opaque crusted ocular discharge (courtesy of Dr. William McRee).

**Figure 2 viruses-16-00508-f002:**
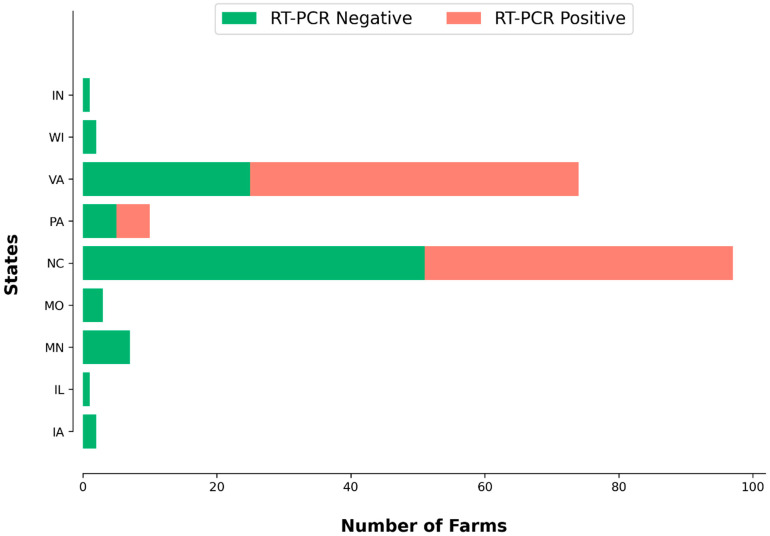
aMPV/B PCR results from turkey farms throughout different US states.

**Figure 3 viruses-16-00508-f003:**
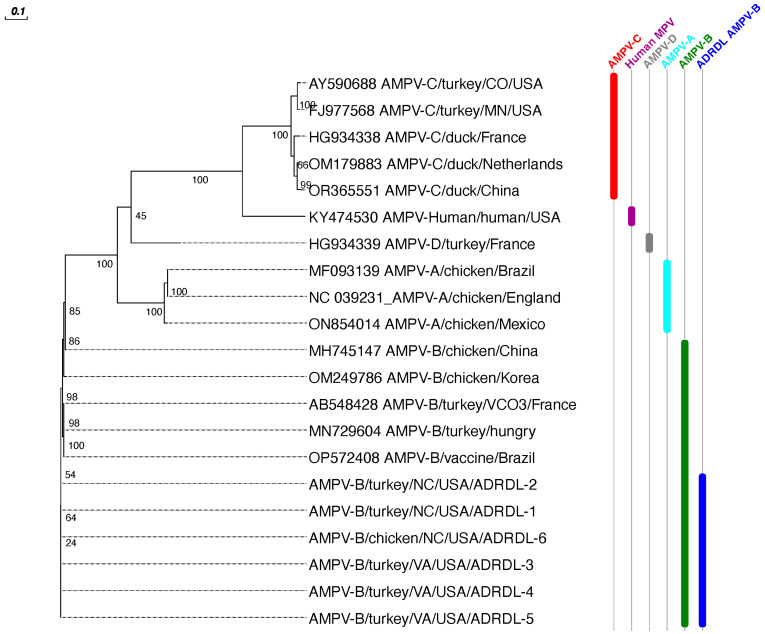
Phylogenetic tree constructed based on whole genome nucleotide sequences of the newly detected subgroup B and the other subgroups A, B, C, and D. The nucleotide sequences tree was generated by maximum likelihood using PhyML with the JC69 substitution model and 100 bootstrap replicates.

**Figure 4 viruses-16-00508-f004:**
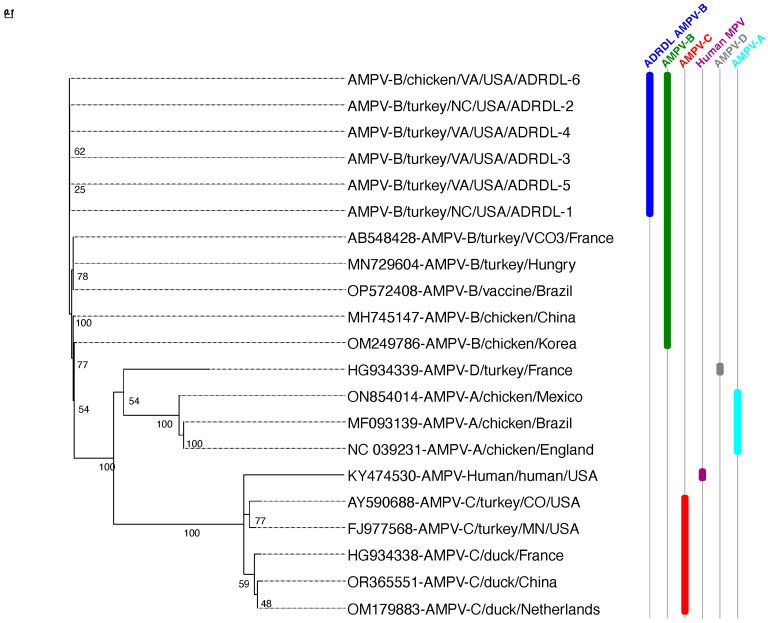
Phylogenetic tree constructed by comparing the aMPV G attachment protein of the newly detected subgroup B and the other subgroups A, B, C, and D. The nucleotide sequence tree was generated by maximum likelihood using PhyML with the JC69 substitution model and 100 bootstrap replicates.

**Figure 5 viruses-16-00508-f005:**
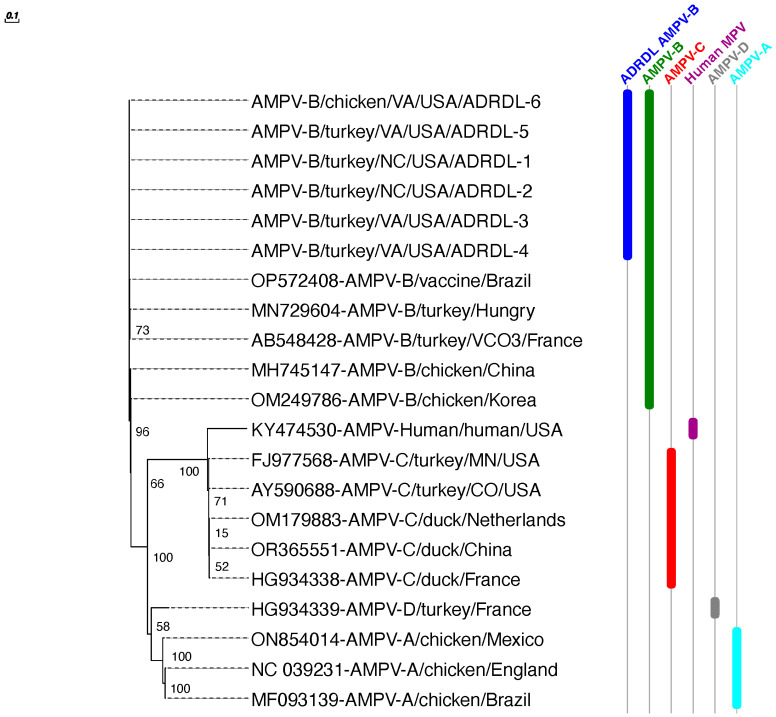
Phylogenetic tree constructed by using aMPV F fusion protein to compare between the newly detected subgroup B and the other subgroups A, B, C, and D. The nucleotide sequences tree was generated by maximum likelihood using PhyML with the JC69 substitution model and 100 bootstrap replicates.

**Figure 6 viruses-16-00508-f006:**
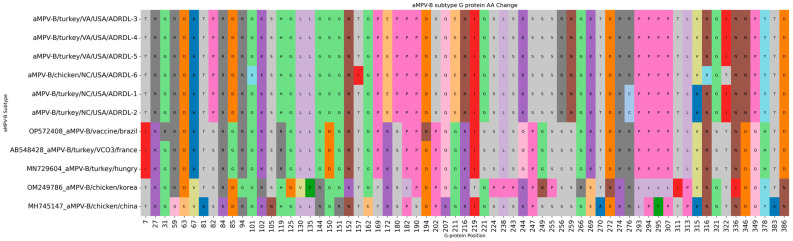
Amino acid substitutions in G protein between the newly detected subgroup B and the other closely related vaccinal and field strains of aMPV/B.

**Table 1 viruses-16-00508-t001:** Representative sequences of aMPV subtypes used in phylogenetic analysis.

Accession Number	Subtype	Country	Species	ID
ON854014.1	A	Mexico	Chicken	aMPV/ck/MEX/3155/22
NC_039231.1	A	UK	Chicken	LAH A
MF093139.1	A	Brazil	Chicken	chicken/Brazil-SP/669/2003
MN729604.1	B	Hungry	Turkey	Hungary/657/4
AB548428.1	B	France	Turkey	VCO3/60616
OP572408.1	B	Brazil	vaccine	aMPV-B/BR/1890/E1/19
OM249787.1	B	Korea	Chicken	chicken/Korea/21004-PLQ7/2021
MH745147.1	B	China	Chicken	LN16
AY590688.1	C	USA	Turkey	Colorado
FJ977568.1	C	USA	Turkey	MN/turkey/2a/97
HG934338.1	C	France	Muscovy duck	1999/99178/
OM179883.1	C	Netherland	Mallard Duck	NL/1/2019
OR365551.1	C	China	Jinding duck	2022/HL1
KY474530.1	HMPV	USA	Human	hMPV/USA/AR002/2016
HG934339.1	D	France	Turkey	Turkey/1985/Fr85.1

**Table 2 viruses-16-00508-t002:** Nucleotide and amino acid identity among the American subtype B and the other subtypes A, B, C, and D based on whole genome sequence (nucleotides only) and on G attachment and F fusion proteins.

	WGS	G Protein		F Protein	
		Nucleotide	Amino Acids	Nucleotide	Amino Acids
aMPV/B	97.74 to 98.58%	93–97%	93–95%	99%	99%
aMPV/A	65–68%	52%	32%	74%	74%
aMPV/C	52–55%	25%	12–15%	67%	67%
hMPV	54%	20%	5%	64%	64%
aMPV/D	65%	39%	25%	72%	72%

## Data Availability

The complete genome sequences of six aMPV-B (five turkey and one chicken) isolates analyzed in this study are publicly available in GenBank under accession numbers PP273456-PP273461.

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
