# Peer review of "Geographical Expansion of Avian Metapneumovirus Subtype B: First Detection and Molecular Characterization of Avian Metapneumovirus Subtype B in US Poultry"

_viruses, 2024, doi:10.3390/v16040508_

Round 1
Reviewer 1 Report
Comments and Suggestions for Authors
The paper “Geographical expansion of avian metapneumovirus subtype B: first detection and molecular characterization of avian metapneumovirus subtype B in US poultry” describes the molecular characteristics of aMPV subtype B detected during disease outbreaks in US poultry farms. This topic is of particular of particular interest for scientific community as it is the first report of this emerging pathogen and related diseases in north American poultry industry.
Following few comments to improve the clarity of the manuscript
MATERIALS AND METHODS
2.1. Clinical cases. How many samples per flock were collected and processed?
2.1 Metagenomic sequencing. It’s not clear the molecular diagnostic process. How many samples were processed for next generation sequencing?
Results
3.3 Realtime RT-PCR. This chapter should be moved after related to “de novo genome assembly and annotation” In my opinion this movements better reflects the consequentiality of the analyses performed
Author Response
Dear Editor,
Thank you for considering our article (viruses-2907986) for publication. We are grateful for the valuable suggestions and comments from the reviewers. We have revised our manuscript in accordance with the reviewers' feedback.
We are thankful to you and reviewers for their valuable time to enhance the quality of the manuscript. All the revised changes are illustrated with track changes in manuscript. Please find below the answers to each specific query.
Response to reviewer 1 comments:
Materials and Methods
Comment 1: Clinical cases. How many samples per flock were collected and processed?
Response 1: Thank you for your input. 5-10 tissue samples or two pools of the swab samples in BHI (11 swabs in each pool) per flock were submitted to the ADRDL-SDSU.
Comment 2: Metagenomic sequencing. It’s not clear the molecular diagnostic process. How many samples were processed for next generation sequencing?
Response 2: Thank you for your careful review. A total of 39 samples were processed by NGS from both turkey and chicken farms.
Results
Comment 3: Realtime RT-PCR. This chapter should be moved after related to “de novo genome assembly and annotation” In my opinion this movements better reflects the consequentiality of the analyses performed.
Response 3: Thank you for your valuable suggestion. We have incorporated the suggested change in the manuscript.
Reviewer 2 Report
Comments and Suggestions for Authors
The information reported by Luqman et al. is of significant interest from both clinical and epidemiological perspectives. The manuscript is overall well-written and clear. The Materials and Methods section and the results are properly reported, and the discussion is appropriate, focusing on the main findings while avoiding overstatements.
I have a few comments and suggestions that could be considered or mentioned in the manuscript to enhance its quality. I am providing an annotated PDF with specific comments.
I suggest a careful revision of the references, as some are not suitable (better references could be found to support the statements) or are incorrectly reported in the reference list (e.g., duplication).

The language used is clear, though some sentences may appear complex for non-native speakers.
Author Response
Dear Editor,
Thank you for considering our article (viruses-2907986) for publication. We are grateful for the valuable suggestions and comments from the reviewers. We have revised our manuscript in accordance with the reviewers' feedback.
We are thankful to you and reviewers for their valuable time to enhance the quality of the manuscript. All the revised changes are illustrated with track changes in manuscript. Please find below the answers to each specific query.
Response to reviewer 2 comments:
Comment 1: Page-3 contains eight
Response 1: Thank you for your careful review, we have corrected the typo.
Comment 2: Page-4-I would not consider this the most appropriate reference. Moreover, the genotype characterization is typically based on the analysis of the genome mutations, rather than AA.
Response 2: We have replaced the reference with a more relevant one. DOI:10.1016/j.vetmic.2010.04.014.
Comment 3: Page-4-Although I agree with the general concept, I believe that stating "any mutation ...has significant effect..." may be to strong.
Response 3: We agree with the reviewer's feedback and have changed the sentence in the manuscript.
Comment 4: Page-4-subtype C was detected also in Europe, see:
"Research Note: Detection of Avian metapneumovirus subgroup C specific antibodies in a mallard flock in Italy"
"Molecular Survey on A, B, C and New Avian Metapneumovirus (aMPV) Subtypes in Wild Birds of Northern-Central Italy"
Response 4: Thank you for your suggestion. We have added Europe in the manuscript.
Comment 5: Page-6-How many samples were processed using this approach? All the samples? just a subset.
Response 5: Initially a subset of samples (n=39 samples) submitted to the ADRDL were processed for NGS for diagnostics. After the confirmation aMPV-B, further samples were screened through real-time RT-PCR.
Comment 6: Page-9-coming of feed to be decrease feed intake.
Response 6: We have corrected the typo in the manuscript.
Comment 7: Page-9-then to be (followed by the appearance).
Response 7: We have corrected the typo in the manuscript.
Comment 8: Page-9-The diagnosis of secondary pathogens is not reported in M&M. How was it performed? on which pathogens? Was it performed by NGS only?
Response 8: We have reported secondary pathogens based on personal communication with field veterinarians and pathologists. Most of samples were tested for bacterial culture and PCR of common respiratory viruses before submitting samples to us for NGS. We have also observed similar results based on metagenomic (NGS) analysis. However, we did not include metagenomic analysis in the manuscript to clearly present the first detection of aMPV-B in the US.
Comment 9: Page-9-How many samples?
Response 9: Six whole genomes of aMPV/B were successfully assembled and partial contigs were assembled from the total (39) samples.
Comment 10: Page-11-Was the derived vaccine, VCO3/50, included in the analysis?
Response 10: Yes, both VCO3/50 and VCO3/60616 have similar identity with study sequences. We selected the pathogenic strain VCO3/60616 as a representative just to compare pathogenic field strains.
Comment 11: Page-13-The concepts of this section are quite redundant and could be shortened.
Response 11: We have revised the paragraph in the manuscript.
Comment 12: Page-13-If I understood correctly, a commercial kit was used (IDEXX-99-56487). How was it updated?
Response 12: Yes, IDEXX PCR kit was used. We have changed the word “updated” with “implemented”.
Comment 13: Page-15-Probably other references could be more adequate.
Response 13: We have replaced it with the more related reference. Doi:10.1186/s13567-020-00817-6.
Comment 14: Page-15-Actually, the evidence that higher Ct values are detected in area where no coexistence with turkey exist, might support that, in this case, turkeys are playing a major role on AMPV-B epidemiology in US.
Response 14: Thank you for the valuable comment. We hypothesize by following this reference,
“These findings lead to the main role of chickens in the maintenance and circulation of aMPV, especially in this area where both species are closely reared. Consequently, the implementation of a prevention strategy in broiler farming that is similar to the one currently adopted in turkeys, appears essential and more than just useful”
Reference DOI: https://doi.org/10.3382/ps/pex350
Comment 15: Page-21-Control the references, several are repeated.
Response 15: Thank you for careful review. We have carefully corrected the references.
Comment 16: Page-31-The bootstrap support and the scale should be reported in all the trees.
Response 16: We have generated and included the phylogenetic trees using 1000-bootstrap. We have updated all trees with bootstrap values.